# The Species Diversity of the Genus *Echinogorgia* in Xiamen Bay and Its New Record in China

**Yun-Pei Wang** [1,2], **Jing Yang** [1,2], **Ta-Jen Chu** [1,2] and **Jia-Ying Liu** [1,2,*]

1 Fisheries College, Jimei University, Xiamen 361021, China; 15605022839@163.com (Y.-P.W.); 18282565232@163.com (J.Y.); chutajen@gmail.com (T.-J.C.)
2 Fujian Provincial Key Laboratory of Marine Fishery Resources and Eco-Environment, Jimei University, Xiamen 361021, China
* Correspondence: jiayingliu@jmu.edu.cn

**Abstract:** The rapid reduction in coral reefs worldwide has led to increasing attention toward protecting and restoring coral reef ecosystems. Coral reefs not only have a rich diversity of coral species, but they can also provide important products and services for human beings. One type of coral, *Echinogorgia*, has important scientific research value and application prospects. To understand the diversity of coral species, diving surveys were conducted in Xiamen Bay in 2017 and 2021, and a total of 928 samples were collected. Taxonomic research was conducted using methods such as morphological identification through electron microscopy. Specific phylogenetic trees of the COI gene, mtMuts gene, and ITS1 gene were analyzed. There were 47 specimens of *Echinogorgia* coral included among 928 samples. Fifteen species of *Echinogorgia* were identified, including *Echinogorgia ramosa*, *Echinogorgia flexilis*, *Echinogorgia russelli*, *Echinogorgia ramulosa*, and *Echinogorgia gracilima* (which represent the newly recorded species in the waters of China). This study increases the species diversity records in China and contributes to new geographical distribution information of *Echinogorgia* worldwide. The primary data also serve as the baseline data for long-term biomonitoring programs to estimate the status of octocorals in Xiamen Bay.

**Keywords:** *Echinogorgia*; new record species; taxonomy; biodiversity; Xiamen Bay

## 1. Introduction

Coral reefs, often referred to as an "oasis in tropical ocean deserts" or "tropical rain forest in the ocean", are known for their ability to provide excellent habitats for numerous organisms; in addition, they form unique and highly productive ecosystems [1]. According to the statistics, coral reefs are distributed in more than 100 countries and territories. Even though they cover less than 0.2% of the seafloor, they support multiple functions, including marine species diversity, coastal protection, food, and economic value [2]. Coral reef ecosystems are estimated to provide 2.85% of the value and services of marine ecosystems [3] and they are considered to have made an inestimable contribution to the world. Since the discovery of recorded corals, the investigation and classification of coral resources have been one of the research focuses of scholars.

Corals are a global biological resource. In terms of distribution, the Indo–Pacific region, including the Pacific Ocean, Southeast Asia Sea, Indian Sea, and Red Sea, accounts for 92% of the world's coral reef area [4]. According to a certain report [5], the second most populous place for coral reefs in Southeast Asia is the South China Sea, with an estimated area of 37,935 square kilometers, which accounts for about 5%. In China, coral reefs also appear in certain sea areas, such as the Fujian, Guangdong, Guangxi, and Hainan coasts [4]. Zou [4] believes that the hermatypic corals community in Dongshan, Fujian is the northernmost sea area where hermatypic corals appear in China.

In China, a number of coral species have been recorded, including 80 genera, or subgenera, and more than 700 species. A total of 1422 species of cnidarians are included in

the "List of Marine Biota of the China Sea" [6]. Among them, there are 763 species of coral, including 41 species of Cerriantipatharia of Anthozoa, 328 species of Octocorollia, and 394 species of Seleactinia of Hexacorollia. As of 2020, China's corals have been recorded in two categories, as well as in sixteen families, seventy-seven genera, and four hundred forty-five species [7]. In the waters of Fujian, there are fifty-four species of gorgonians in twenty-two genera and nine families [8]. The places where Scleractinian corals are distributed in Fujian include Xiamen, Dongshan, and Niushan Island, as well as in the Taishan Islands and other such islands [8,9]. However, there are currently few studies on the coral classification in Xiamen Bay. In 2006, a total of 38 species of coral were recorded in Xiamen Bay. Among them, there were twenty-eight species in seven families of Alcyonacea, and ten species in six families of Scleractinia [10]. In 2007, Huang et al. [11] reported on the species of Octocorallia corals in the waters near Wuyu Island, and these belonged to four families and seven genera. In 2017, Ni et al. [12] found six species of scleractinian corals in Baiha Reef, Xiamen Bay. In 2022, Liu et al. [13] reported three species of *Astrogorgia* coral inhabiting Xiamen Bay. In 2023, Yang et al. [14] found a newly recorded species in Xiamen Bay.

The functions of coral reefs support marine biodiversity and provide important products and services to humans [15,16]. Octocorallia includes blue corals, soft corals, sea pens, and gorgonians (sea fans and sea whips) within three orders: Alcyonacea, Helioporacea, and Pennatulacea [17]. There are two orders and seventy-nine families in Octocorallia [18]. Octocorallia coral plays an important role in maintaining the structural integrity and biodiversity of marine ecosystems [19]. The genus *Echinogorgia* belongs to the Malacalcyonacea of Octocorallia, which is similar to the Paramuriceidae. *Echinogorgia* was established in 1865, and the type species are *E. sasappo* (Esper, 1794), *E. reticulata* (Esper, 1791), and *E. umbratica* (Esper, 1791) [20]. This genus has the following characteristics: the colonies surface is flat, it is in a network-shaped fan shape, and it has short lateral branches on the main branch [20]. Moreover, the sclerites are spindle-shaped and it has leaf-shaped sclerites in the coenenchymata. These corals are red, brown, yellow, or white. As of June 2023, WORMS (https://www.marinespecies.org, accessed on 2 March 2023) records 42 valid species of this genus. The species of this genus occur in many sea areas, including the offshore waters of mainland China, Hong Kong, Germany, and the Indian Ocean [21]. Among them, eight species of this genus have been observed in the waters of China, namely *E. pseudosassapo*, *E. ridley* [22], *E. sassapo*, *E. flora*, *E. furfuracea*, *E. complexa*, *E. aurantiaca*, and *E. lami* [8,11,23]. Among them, *E. ridley* is recorded in the offshore waters of Taiwan, *E. pseudosassapo* is recorded in the offshore waters of Taiwan and Fujian, while the other six species are mainly distributed in the offshore waters of Fujian.

At present, the following two issues are of concern to the genus *Echinogorgia*: one is taxonomic identification and the other is natural active substances. The issue of natural active substances includes the extraction of substances [24,25], as well as the structure and biological activity of substances [26–28]. Most of the above studies show the status of unspecified species [29,30], and this uncertainty is not conducive to future research and production applications. Uncertainty in species classification is mainly due to the fact that previous authors did not assign the holotype and that the specimens and sclerites are mostly insufficient in correctly identifying the species. Furthermore, certain species have been identified based on only a few specimens or fragments [31]. Therefore, an in-depth taxonomic study of the species in this genus is crucial.

With the rapid development of molecular technology, these tools are also widely used in coral classification and identification. Certain studies have shown that 18S rDNA [32,33], ITS [34,35], COI, ND2, and mtMuts gene fragments [36–38] are suitable for the phylogenetic study of corals. On the basis of morphological identification, Xu et al. [39] established a phylogenetic tree of the mtMuts gene to study the taxonomy and phylogeny of Octocorallia in western Pacific seamounts. Li et al. [33] clearly demonstrated the phylogenetic relationship of Gorgonian coral by establishing a 18S rDNA phylogenetic tree. McFadden et al. [18] used the mtMuts gene to establish a phylogenetic tree and conducted a phylogenetic analysis on

185 taxa of Octocorallia coral. Hume et al. [40] studied coral species diversity using ITS2 markers.

In this study, the morphological classification method was used to identify the samples of *Echinogorgia* collected from Xiamen Bay. Also, via molecular techniques, phylogenetic analysis was conducted using the COI gene sequences, mtMuts gene sequences, and ITS gene sequences. The determination of the specific types of corals in Xiamen Bay will help with providing data support for biodiversity conservation in Xiamen Bay.

## 2. Materials and Methods

### 2.1. Station and Sample Collection

Nine hundred and twenty-eight coral samples were collected via diving sampling in 2017 and 2021. Among them, 823 samples were collected in 10 stations in 2017. In 2021, 105 samples were collected from 4 stations. The information collected from the sampling stations is shown in Table 1, and the schematic diagram of the sampling stations is shown in Figure 1.

**Table 1.** The sampling station information from 2017 and 2021.

| Year | Station | Longitude | Latitude |
|---|---|---|---|
| 2017 | Shangyu Island (SY) | 118°11′19″–118°11′25″ | 24°27′12″–24°27′13″ |
| | Huangcuo (HC) | 118°07′50″ | 24°25′25″ |
| | Kulangsu Island (KLS) | 118°03′14″–118°03′41″ | 24°26′23″–24°26′35″ |
| | Fire Island (FI) | 118°03′54″ | 24°29′35″ |
| | Qingyu Island (QY) | 118°05′35″–118° 07′28″ | 24° 21′45″–24°21′55″ |
| | Wuyu Island (WY) | 118° 03′54″–118° 08′35″ | 24° 20′23″–24° 29′35″ |
| | Dabai Island (DB) | 118°26′58″–118°27′40″ | 24°33′46″–24°34′9″ |
| | Xiaobai Island (XB) | 118°27′47″ | 24°33′21″ |
| | Jiaoyu Island (JY) | 118°24′14″ | 24°32′41″ |
| | Baiha Reef (BH) | 118°22′07″–118°22′17″ | 24°31′38″–24°31′55″ |
| 2021 | Qingyu Island (QY) | 118°07′21″–118°07′50″ | 24°21′45″–24°22′11″ |
| | Wuyu Island (WY) | 118°08′28″–118°08′57″ | 24°20′31″–24°20′51″ |
| | Dabai Island (DB) | 118°45′02″–118°46′06″ | 24°56′31″–24°57′51″ |
| | Xiaobai Island (XB) | 118°42′58″–118°43′38″ | 24°58′26″–24°58′96″ |

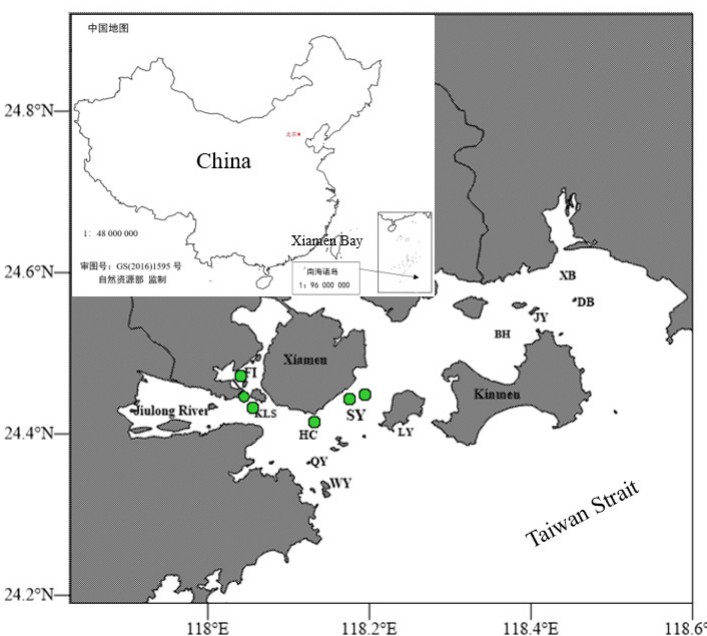

**Figure 1.** Schematic diagram of sampling stations.

*2.2. Material and Sample Processing*

The samples were collected and brought back to the laboratory in order to take photos of the living corals, and these were numbered for preservation. Suitable lengths of coral samples were cut and placed in centrifuge tubes filled with 95% ethanol for traditional morphological and molecular biology experiments. The remaining samples were stored in an ultralow temperature refrigerator at −70 °C.

*2.3. Experimental Instruments*

In this study, certain experimental instruments were used, including a Nikon SMZ1270 dissecting microscope, a NiKon ECLIPSE 80i microscope, a DHG-9050A air-drying oven, a G5 PhenomProX2017 electron scanning electron microscope, an A200 PCR amplification apparatus, a Z216MK centrifuge, an MK200-2 dry thermostat, a DYY-8C electrophoresis apparatus, and a JS-20.2 gel imaging system.

*2.4. Identification Method*

Generally, octocorals are identified, classified, and described primarily on the basis of external and internal morphology. These morphological characteristics include the colonies' size, shape, color, and calyx structures, as well as their sclerite content, dominance, shape, size, and arrangement [31,32,37,41].

(1) Exterior feature observation

First, certain characteristics were observed, including the coral appearances, colors, polyps, surfaces, spindles, branches, bases, top colonies, etc. In addition, actions were carried out, including measuring, photographing, describing, and recording the relevant parts of the coral.

(2) Observation of local appearance and morphology

A dissecting microscope (Nikon SMZ1270) was used to observe (as well as photograph) the color and shape of the polyps, as well as the calyx structure.

(3) Observation of sclerites

After the tissue samples were dissolved in sodium hypochlorite, different parts of the sclerite samples (polyps, coenenchymata, and tentacles) were observed and examined. The sclerites were washed with distilled water and subsequently dried using a blast dryer. The samples were sent to the scanning electron microscope (SEM) (G5 PhenomProX2017) set up in the National Key Laboratory of Xiamen University for observation. Then, detailed images of the sclerites through SEM were obtained. The images for each sclerite type were then optimized using Adobe Photoshop 2020. When the coral branches were placed in the Petri dish, a solution of ultra-pure water and sodium hypochlorite (with a ratio of 3:1–5:1) was injected into it for dissolution. Next, the arrangement of the spicules under a dissecting microscope were observed and pictures were taken.

*2.5. Classification Basis*

2.5.1. Identification Documents

Certain important documents were used for species classification and identification such as "Soft Corals and Sea Fans" [42], the World Register of Marine Species (WROMS) [21], and taxonomic terminology references [43].

2.5.2. DNA Extraction and PCR Amplification

With the use of the marine animal tissue genomic DNA extraction kit of Shanghai Biotechnology Co., Ltd., three coral individuals were prepared for total genome DNA extraction. The steps of extraction were followed as per the manufacturer's instructions. The primers used in this study are shown in Table 2.

**Table 2.** The primers used for genetic analyses.

| Primers | Gene Region | Sequence (5′-3′) | References |
|---|---|---|---|
| COI-LA-8398 | COI gene | F-GGA ATG GCG GGG ACA GCT TCG AGT ATG TTA ATA CGG | [44] |
| COIoct | | R-ATC ATA GCA TAG ACC ATACC | |
| AnthoCorMSH | MSH gene | F-AGG AGA ATT ATT CTA AGT ATGG | [44] |
| Mut3458R | | R-TSG AGC AAA AGC CAC TCC | |
| ITS1 | ITS gene | F-TCCGTAGGTGAACCTGCGG | [45] |
| ITS2 | | R-GCTGCGTTCTTCATCGATGC | |

2.5.3. Phylogenetic Analysis of Coral Samples Based on COI, mtMuts, and ITS Gene Fragments

Based on the morphological results, using molecular technology to extract the DNA from certain coral samples for amplification analysis, the COI gene sequence, mtMuts gene sequence, and ITS1 gene sequence results of the samples were compared with the GenBank database. The gene sequence fragments of the species with a similarity of over 95% were compared with the corresponding screened gene fragment information in the NCBI database using MEGA7.0 software. The neighbor joining method, based on the Kimura 2 parameter (K2P) model with 1000 bootstrap replicates, was used to construct phylogenetic trees and to conduct a phylogenetic analysis.

## 3. Results

### 3.1. Systematic Approaches

The morphological classification method was used to identify 47 samples of the genus *Echinogorgia*, and the results showed that these samples were classified into 15 species. A total of 15 species named *Echinogorgia ramosa*, *E. flexilis*, *E. russelli*, *E. ramulosa*, *E. gracillima*, *E.* sp1, *E.* sp2, *E.* sp3, *E.* sp4, *E.* sp5, *E.* sp6, *E.* sp7, *E.* sp8, *E.* sp9, and *E.* sp10 were identified. We found five new recorded species in China, including *E. ramose*, *E. flexilis*, *E. russelli*, *E. ramulosa*, and *E. gracillima*.

The morphological descriptions of the five newly recorded species were as follows.

3.1.1. *Echinogorgia ramosa* (Thomson and Henderson, 1905)

Sample collection location: Qingyu (118°07′34″, 24°21′46″), water depth: 7.5 m;
Study sample number: 20210423-QY-33;
Coral group (Figure 2a): The living colonies were 9 cm high, and most of their branches were on the same plane. The lateral branches bent outward and upward. The irregular branches were mainly distributed on one side of the main trunk. The diameter of the central shaft was greater than 0.5 mm. The central axis was divided into two layers: inner and outer (with an uneven yellow color on the inner layer and a uniform dark brown color on the outer layer). The ratio of the inner and outer layers of the axis radius was approximately 2:3;
Polyps (Figure 2a(B,C)): The polyps could freely contract and were mainly distributed on the surface of the trunk and branches. The polyps contracted into the coral calyx, which was circular in shape and had a diameter of about 0.5 mm;
Sclerites (Figure 2b): The polyps contained straight or curved spindle-shaped sclerites with a length of about 0.2–0.4 mm. The surface of the sclerites contained small and conical columnar protrusions. The coenenchymata contained leaf-shaped sclerites and a few pan-shaped sclerites with a length of about 0.1–0.25 mm. The leaf-shaped sclerites contained serrated edges, and the surface of the sclerites presented protrusions of varying sizes. Sclerites are colorless;
Color (Figure 2a(A)): reddish brown, and dark brown in alcohol;
Sample collection location: the waters near Qingyu Island in Xiamen Bay.

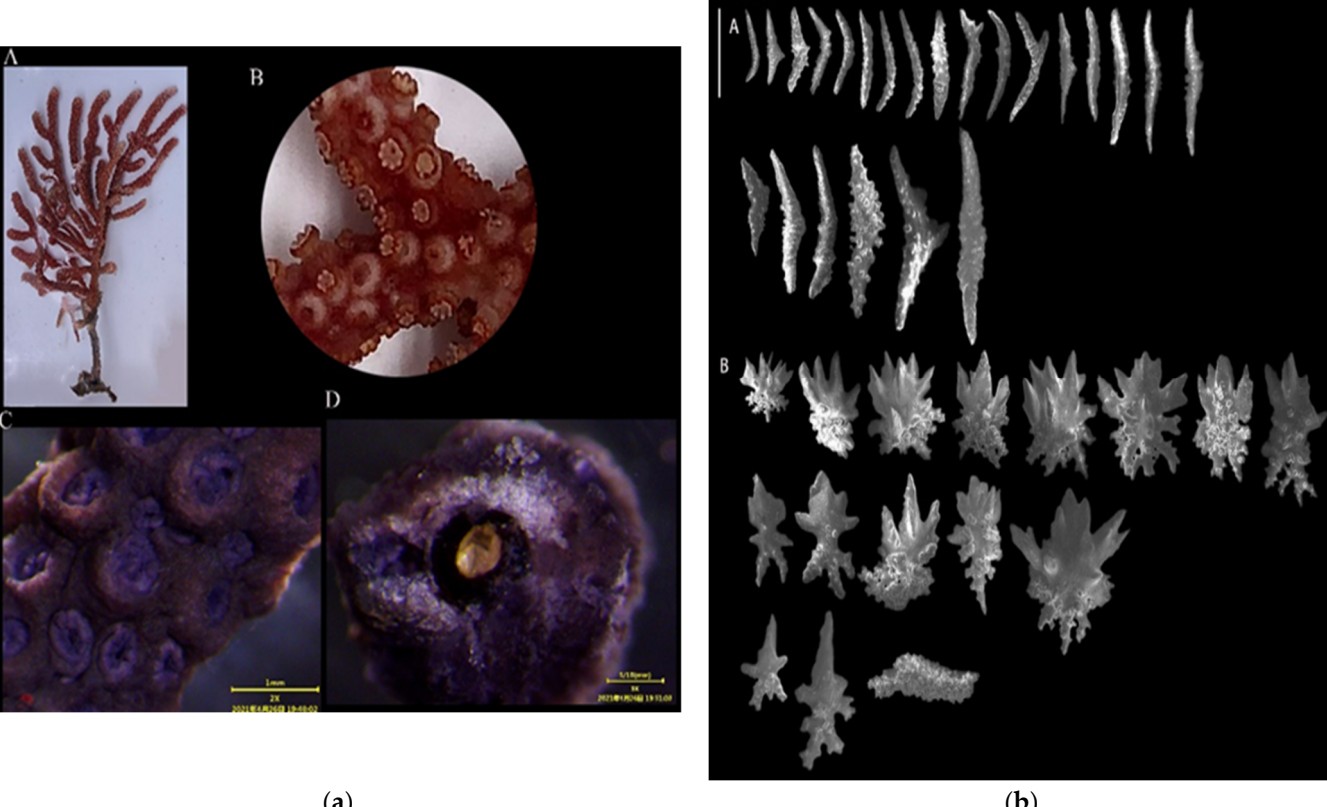

**Figure 2.** *E. ramosa*: (**a**) Corals' external morphologies ((**A**). Coral colonies; (**B**,**C**). Polyps; (**D**). Axis); (**b**) SEM images of the coral sclerites ((**A**). Sclerites of the polyps; (**B**). Sclerites of the coenenchymata; Scales: A = B = 0.2 mm).

3.1.2. *Echinogorgia flexilis* (Thomson and Simpson, 1909)

Coral group (Figure 3a): The living colonies were in a fan-shaped shape and were up to 10 cm high. The branches were irregular and on the same plane. The central axis had two layers. The inner layer was an uneven yellow, and the outer layer was a uniform dark brown. The axis radius was 3:2;

Polyps (Figure 3a(B,C)): The polyps' monotypes were distributed on the surface of the stems and branches. The gaps between the polyps were approximately 0.5 mm in diameter. The coral calyx protruded and appeared as a dome, and the polyps contracted into the calyx. The polyps turned purple after alcohol immersion;

Sclerites (Figure 3b): Straight or slightly curved fusiform-shaped sclerites were distributed in the polyps and were about 0.18–0.3 mm long, and some of the fusiform-shaped sclerites were bifurcated at one end with columnar and prickly protrusions on the sclerites. The surface layers of the coenenchymata had leaf-shaped, flat-shaped, and irregular-shaped sclerites with a length of about 0.1–0.35 mm. The leaf-shaped sclerites had serrated edges, and the surface of the sclerites all contained warty protrusions. The inner layers of the coenenchymata contained multiple spicules (approximately 0.05–0.15 mm long), and the sclerites were colorless;

Color (Figure 3a(A)): brownish red, and dark brown in alcohol;

Sample collection location: the waters near Qingyu Island in Xiamen Bay.

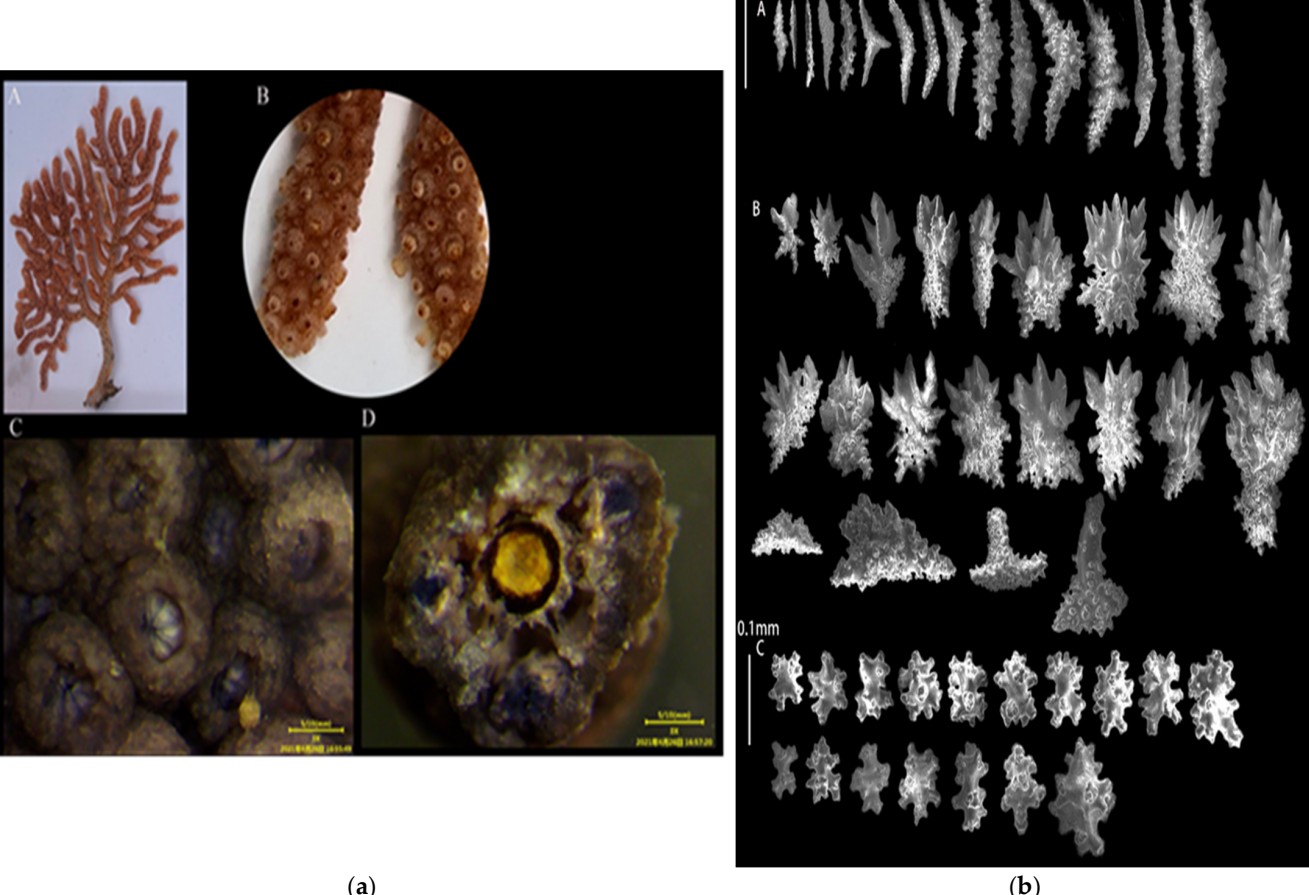

**Figure 3.** *E. flexilis*: (**a**) The corals' external morphologies ((**A**). Coral colonies; (**B**,**C**). Polyps; (**D**). Axis); (**b**) SEM images of the coral sclerites ((**A**). Sclerites of the polyps; (**B**). Sclerites of the surface layers of coenenchymata; (**C**). Sclerites of the inner layers of coenenchymata; Scales: A = B = 0.2 mm; C = 0.1 mm).

### 3.1.3. *Echinogorgia russelli* (Bayer, 1949)

　　Sample collection location: Wuyu (118°8′57″,24°2′51″), 5 m;

　　Study sample number: 20210423-WY-15;

　　Coral group (Figure 4a): The living colonies were 20 cm tall with dense and irregular branches that bent upward and downward. Most of the branches were at right angles with a small number of the coenenchymata shedding from the base of the main stem. The diameter of the central axis was about 1 mm and was an uneven yellow color;

　　Polyps (Figure 4a(A,B)): The polyps could contract completely and freely, and they were mainly distributed on the surface of the main stems and branches. The calyx slightly protruded, and it resembled a low dome-shaped wart. The polyps completely contracted into the calyx with a diameter of less than 1 mm. After alcohol immersion, the polyps turned purple;

　　Sclerites (Figure 4b): The polyps contained columnar- and spindle-shaped sclerites, about 0.18–0.3 mm long, with columnar and spiny processes on the sclerites. The outer layers of the coenenchymata contained leaf-shaped sclerites, approximately 0.18–0.25 mm long, with serrated edges on the sclerites. The inner layers of the coenenchymata contained quadrangular sclerites with a length of approximately 0.05–0.08 mm. The sclerites were colorless;

　　Color (Figure 4a(A)): light brownish red, and dark brown in alcohol;

　　Sample collection location: the waters near Wuyu Island in Xiamen Bay.

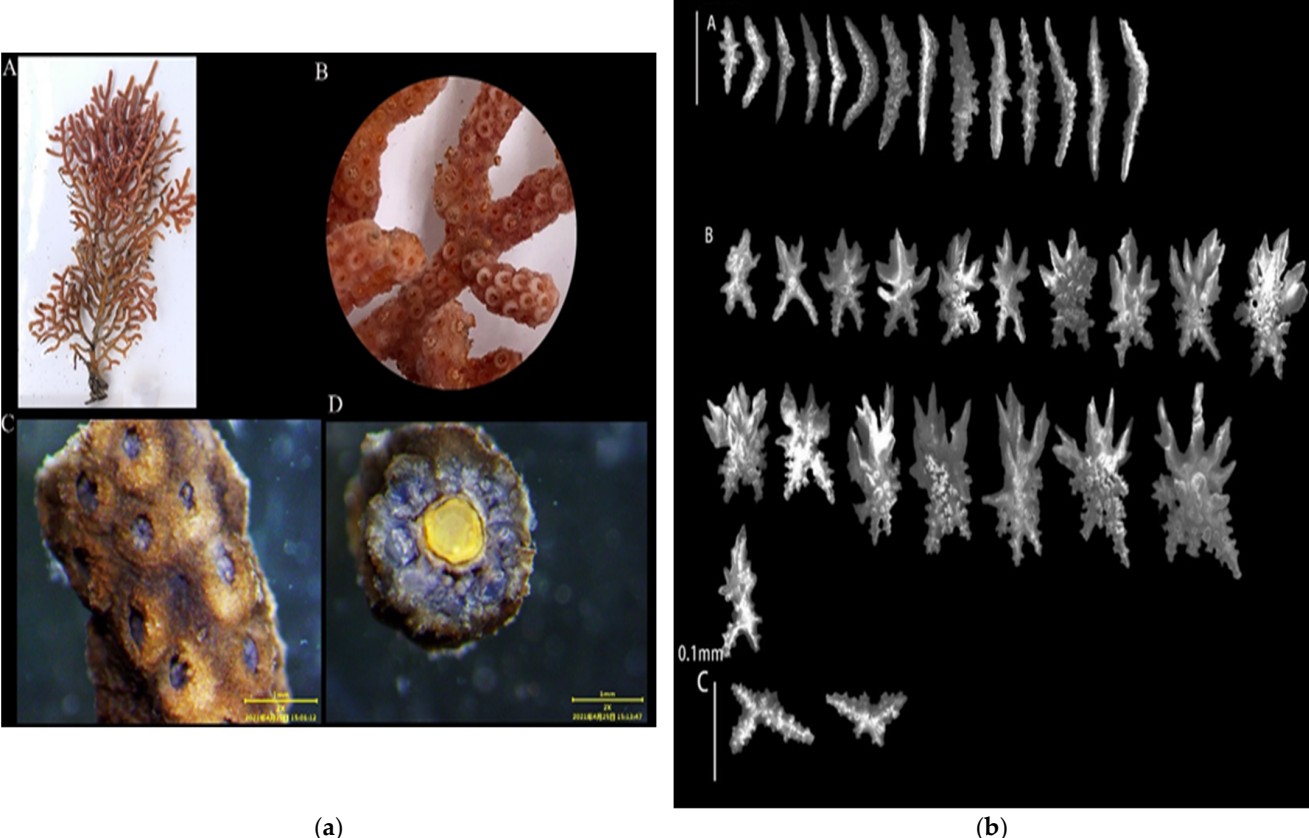

(**a**) (**b**)

**Figure 4.** *E. russelli*: (**a**) The corals' external morphologies ((**A**). Coral colonies; (**B**,**C**). Polyps; (**D**). Axis);
(**b**) SEM images of the coral sclerites ((**A**). Sclerites of the polyps; (**B**). Sclerites of the outer layers of
coenenchymata; (**C**). Sclerites of the inner layers of coenenchymata; Scales: A = B = 0.2 mm; C = 0.1 mm).

3.1.4. *Echinogorgia ramulosa* (Gray, 1870)

Sample collection location: Wuyu (118°8′57″, 24°2′51″), 5 m;

Study sample number: 20210423-WY-15;

Coral group (Figure 5a): The living colonies were branching and were up to 14 cm
high with abundant short branches. The small branches appeared at right angles to the
main branch, and the short branches were slightly swollen at the end. The coenenchymata
were thicker, and the diameter of the central axis was less than 0.5 mm. The central axis
was uniformly yellow;

Polyps (Figure 5a(B,C)): The polyps appeared uniformly on the surface of the trunks and
branches, with a diameter of about 0.5 mm. The calyx protruded, resembling a circle, and some
of the polyps contracted into the calyx. After soaking in alcohol, the polyp turned purple;

Sclerites (Figure 5b): The polyps contained columnar- and spindle-shaped sclerites
about 0.1–0.4 mm long. The spindle-shaped sclerites were relatively flat, and the sur-
face of the spicules contained more verrucous protrusions. The surface layers of the
coenenchymata contained leaf-shaped sclerites and irregular sclerites with a length of
about 0.1–0.2 mm, and the edges of the sclerites contained serrated protrusions. The inner
layers of the coenenchymata contained multiple spicules with a length of approximately
0.1–0.15 mm. The sclerites were colorless;

Color (Figure 5a(A)): brick red, and dark brown in alcohol;

Sample collection location: the waters near Wuyu Island in Xiamen Bay.

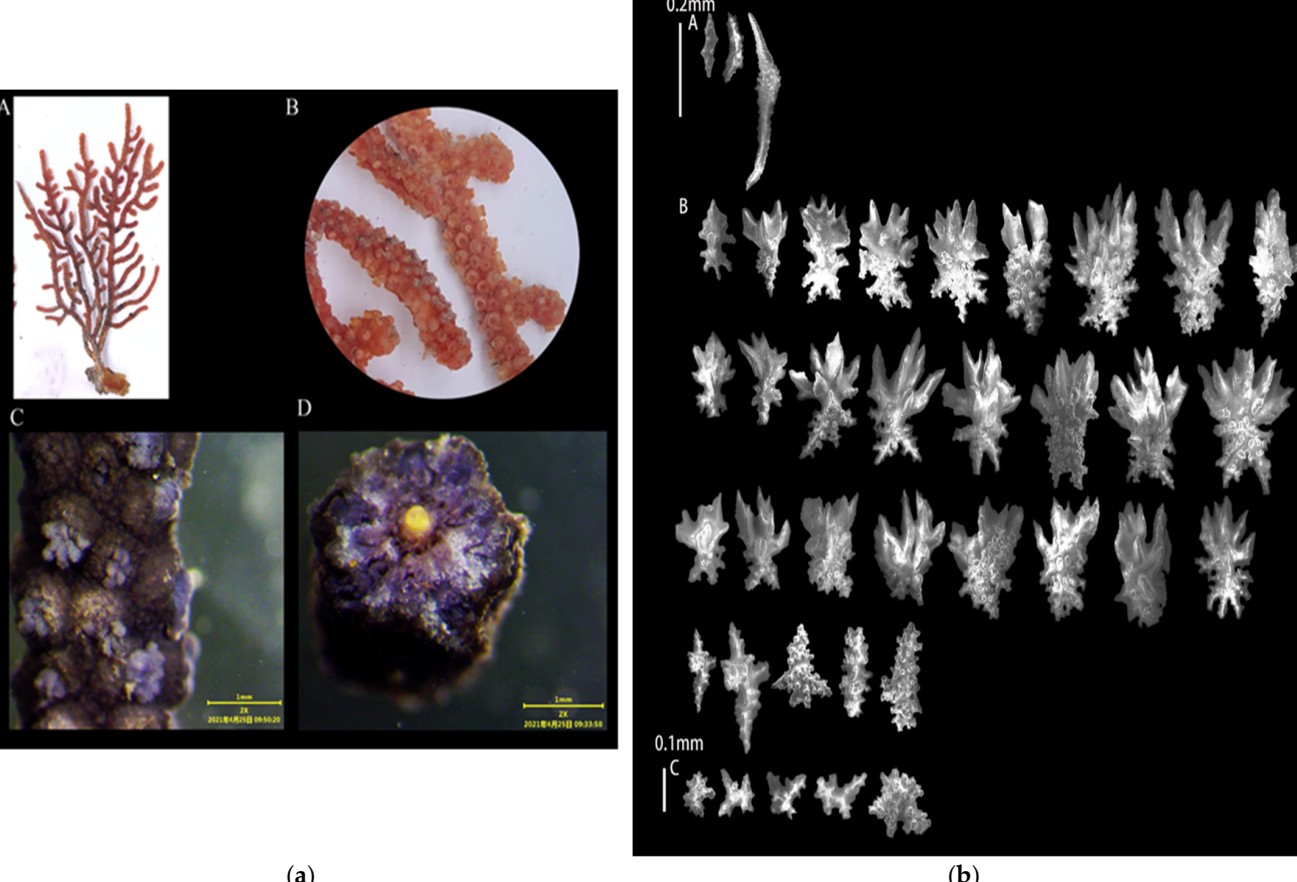

(**a**) (**b**)

**Figure 5.** *E. ramulosa*: (**a**) Corals' external morphologies ((**A**). Coral colonies; (**B**,**C**). Polyps; (**D**). Axis); (**b**) SEM images of the coral sclerites ((**A**). Sclerites of the polyps; (**B**). Sclerites of the outer layers of coenenchymata; (**C**). Sclerites of the inner layers of coenenchymata; Scales: A = B = 0.2 mm; C = 0.1 mm).

### 3.1.5. *Echinogorgia gracillima* (Kükenthal, 1917)

Sample collection location: Wuyu (118°8′57″, 24°2′51″), 5 m;

Study sample number: 20210423-WY-11;

Coral group (Figure 6a): The living colonies were fan-shaped and were up to 10 cm with abundant and overlapping branches. There were a large number of lateral branches on one plane with slightly enlarged ends. The inner layer of the central axis was uneven and orange–yellow, and the outer layer was an uneven dark brown. The radius of the axis was about 2:1 for the inner and outer layers;

Polyps (Figure 6a(B,C)): The polyps appeared on the surface of the trunks and branches with protruding calyxes. The polyps could fully contract into the calyx with a diameter of about 0.5 mm. After alcohol immersion, the polyps turned yellow;

Sclerites (Figure 6b): The polyps contained columnar- and spindle-shaped sclerites, which were about 0.1–0.25 mm long. The sclerites contained verrucous protrusions, and some of the spindle-shaped sclerites were bifurcated at one end. The outer layers of the coenenchymata contained leaf-shaped sclerites, approximately 0.15–0.2 mm long, with serrated protrusions at the edges of the sclerites. The inner layers of the coenenchymata contained radiating sclerites, approximately 0.03–0.04 mm in length. The sclerites were colorless.

Color (Figure 6a(A)): yellow, and gray-brown in alcohol.

Sample collection location: the waters near Wuyu Island in Xiamen Bay.

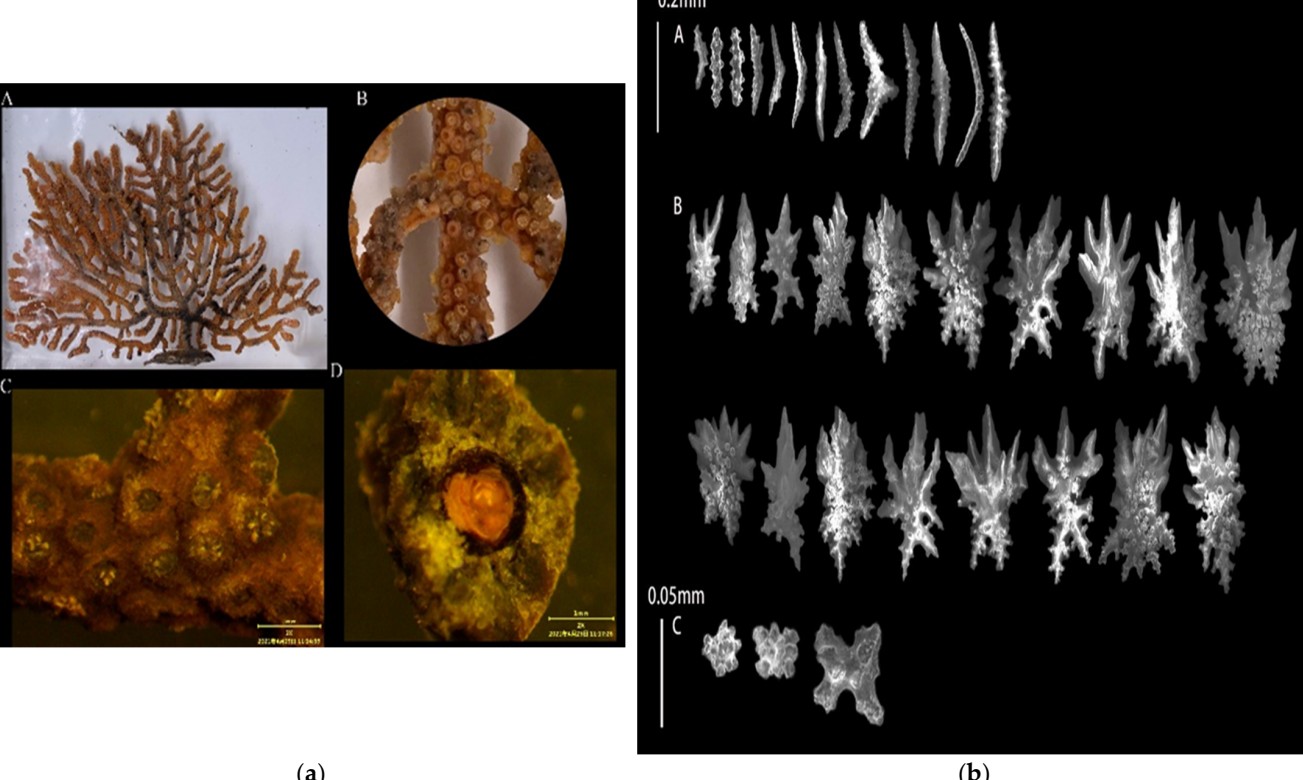

(**a**)                                    (**b**)

**Figure 6.** *E. gracillima*: (**a**) Corals' external morphologies ((**A**). Coral colonies; (**B**,**C**). Polyps; (**D**). Axis); (**b**) SEM images of the coral sclerites ((**A**). Sclerites of the polyps; (**B**). Sclerites of the outer layers of coenenchymata; (**C**). Sclerites of the inner layers of coenenchymata; Scales: A = B = 0.2 mm; C = 0.1 mm).

### 3.2. DNA Analysis

The gene sequence of the successfully amplified *Echinogorgia* coral sample was uploaded to the NCBI database to obtain the gene sequence number, as shown in Appendix A. The gene sequences of the samples were compared with and screened using NCBI's Nucleotide BLAST software (https://blast.ncbi.nlm.nih.gov/Blast.cgi?PROGRAM=blastn& PAGE_TYPE=BlastSearch&LINK_LOC=blasthome, accessed on 12 March 2023), and the corresponding gene fragment information screened in the NCBI database was used in the MEGA7.0 software. The K2P model and NJ method were used to model the phylogenetic trees of the COI gene, mtMuts gene, and ITS1 gene, respectively. The confidence level was derived from 1000 non-parametric bootstrap analyses. The specific phylogenetic trees of the COI gene, mtMuts gene, and ITS1 gene depicted the evolutionary lineages of the different species. The results of the phylogenetic trees are shown in Figure 7 (the login numbers in parentheses in the figure represent the samples of this study).

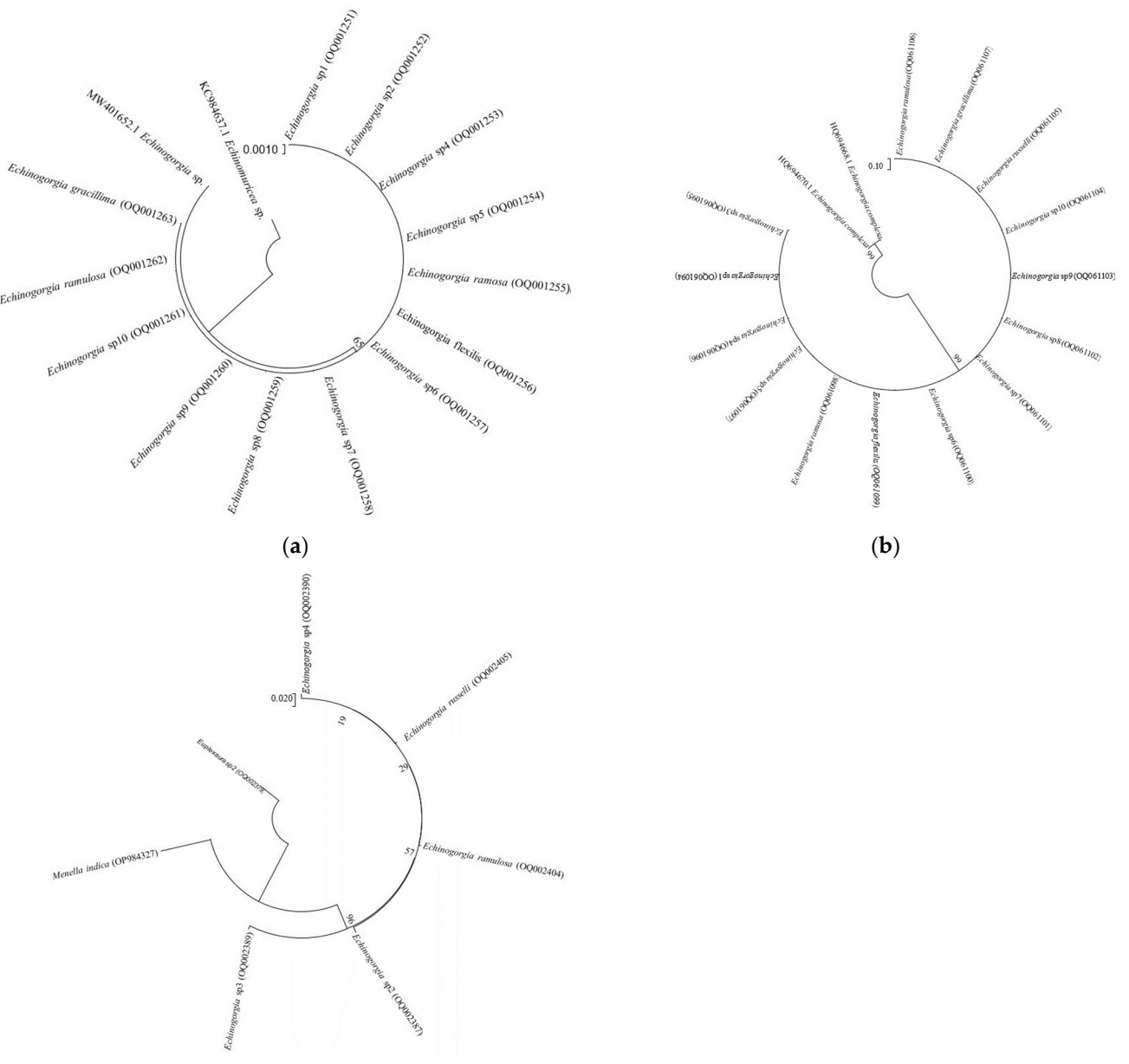

**Figure 7.** NJ phylogenetic tree: (**a**) Based on fragments of the COI gene sequences; (**b**) Based on fragments of the mtMuts gene sequence; (**c**) Based on fragments of the ITS1 gene sequences.

Figure 7a depicts a phylogenetic tree that was established using COI gene sequences. From the figure, it can be concluded that the COI gene sequence can be used to distinguish the class of the genus *Echinogorgia* from the other genus. Our molecular phylogenetic analysis showed that *E.* sp1, *E.* sp2, *E.* sp4, *E.* sp5, *E. ramosa*, *E. flexilis*, *E.* sp6, *E.* sp7, *E.* sp8, *E.* sp9, *E.* sp10, *E. ramulosa*, and *E. gracillima* are genetically closer.

Figure 7b depicts a phylogenetic tree established using mtMuts gene sequences. From the figure, it can be concluded that the mtMuts gene sequence can be used to distinguish the class of the genus *Echinogorgia* from the other genus. Our molecular phylogenetic analysis showed that *E. ramulosa*, *E. gracilima*, *E. russelli*, *E.* sp10, *E.* sp9, *E.* sp8, *E.* sp7, *E.* sp6, *E. flexilis*, *E. ramosa*, *E.* sp5, *E.* sp4, *E.* sp1, and *E.* sp3 are genetically closer.

Figure 7c is a phylogenetic tree established based on the ITS1 gene sequence. From the figure, it can be concluded that the ITS gene sequence can be used to distinguish the

species of the genus *Echinogorgia*. Our molecular phylogenetic analysis showed that *E.* sp4, *E. russelli*, *E. ramulosa*, *E.* sp2, and *E.* sp3 are genetically closer.

## 4. Discussion

*4.1. Key Points for the Morphological Identification of the Genus Echinogorgia*

The external characteristics are the main basis for the classification and identification of Octocorallia corals. The characteristics—such as coral specimens (color, branching, size, and the distribution of the coral), polyps (shape during relaxation and contraction), and sclerites (shape, size, and color)—were observed [22,31,32,41,46]. The living coral sample with the sample number of 20210423-QY-33 was found to be reddish-brown. Most of the branches were on the same plane. There were a few anastomoses between the branches. The branches were irregular and mainly distributed on one side of the main trunk. The polyps contained straight or curved spindle-shaped sclerites, and the coenenchymata contained leaf-shaped sclerites and a small number of pan-shaped sclerites. In the original literature of the species of *Econogorgia ramosa* (Thomson and Henderson, 1905), it was described that the coral body contains a large number of branches, some of which are anastomosed. In addition, the polyps contain spindle-shaped sclerites, the coenenchymata contain leaf-shaped sclerites and pan-shaped sclerites, and the color of the coral body is brick-red [47]. After comparing the morphological characteristics of the coral sample with the original literature description and image of *E. ramosa* (Thomson and Henderson, 1905), it was identified as *E. ramosa* (Thomson and Henderson, 1905).

The coral sample with the sample number 20210423-QY-27 had a fan-shaped coral body, branching coral bodies, and small gaps between the polyps. The calyx protruded and appeared in a dome-like shape. The polyps were distributed with straight or slightly spindle-shaped sclerites. The surface of the coenenchymata contained leaf-shaped, pan-shaped, and irregular-shaped sclerites. The leaf-shaped sclerites contained serrated edges, and the surface of the sclerites contained warty projections. The inner layer of the coenenchymata contained multiple sclerites. The original literature on the species described that the coral body is fan-shaped, the coral body is branched, the calyx is prominent, the polyps are distributed with spindle-shaped sclerites, and the symphysis contains multiple sclerites in *Econogorgia flexilis* (Thomson and Simpson, 1909) [48]. The morphological characteristics of the coral body with the sample number 20210423-QY-27 corresponded to the original literature, and it was identified as the species of *E. flexilis* (Thomson and Simpson, 1909).

The sample number was found to be 20210423-WY-15. The living colonies exhibited dense branching with anastomoses between the branches. The branches were irregular and grew upward and downward. Most of the branches were at right angles. The polyps contained columnar- and spindle-shaped sclerites, and the outer layer of the common flesh contained leaf-shaped bone needles. The coenenchymata in the inner layers contained quadrangular sclerites. The morphological description of its coral body, branches, and the sclerites of polyps corresponded to that described in the original literature of *Econogorgia ruselli* (Bayer, 1949). Its coral body was found to be upright, and the coral body had many branches with many anastomoses between the branches and curved branches; in addition, many of the branches were at right angles. The polyps contained spindle-shaped sclerites, and the coenenchymata contained multi-shoot-shaped sclerites [47]. Therefore, the sample with sample number 20210423-WY-15 was identified as *E. russelli* (Bayer, 1949).

The sample with the sample number 20210423-WY-6 was found to have a branching coral body with abundant short branches. The small branches were at right angles to the main branch, and the end of the short branch was slightly enlarged. The calyx protruded, resembling a circle, and the polyps contracted into the calyx. The polyps contained columnar- and spindle-shaped sclerites that were about 0.1–0.4 mm long. They also had flat spindle-shaped sclerites, and the surface of the sclerites contained more verrucous protrusions. The surface layers of the coenenchymata contained leaf-shaped and irregular-shaped sclerites with a length of approximately 0.1–0.2 mm. The inner layers of the coenenchymata contained multiple spicules with a length of approximately 0.1–0.15 mm. In the original

literature of *Echinogorgia ramulosa* (Gray, 1870), the coral was described as having a branching shape, rich branches, as well as especially short and small branches with right angles between the branches and ends. The polyps are distributed on the surface of twigs. The calyx protrudes. The polyps contain spindle-shaped sclerites with a length of 0.08–0.3 mm. The surface of the sclerites contains many warty projections [49]. By comparing the morphological description of sample number 20210423-WY-6 with the description of the original literature on *E. ramulosa* (Gray, 1870) [50], it was identified as *E. ramulosa* (Gray, 1870) [50].

Sample number 20210423-WY-11 showed a fan-shaped coral body with a height of 10 cm and had abundant and overlapping branches. There were a large number of lateral branches on one plane, and the ends of the branches were slightly enlarged. The polyps had a diameter of about 0.5 mm and protruding coral calyxes. The polyps contained columnar- and spindle-shaped sclerites, which were about 0.1–0.25 mm long. The sclerites contained verrucous protrusions, and some spindle-shaped sclerites were bifurcated at one end. The leaf-shaped sclerites were shown on the outer layers of the coenenchymata, while the inner layers of the coenenchymata contained radiating sclerites, which were colorless. The coral living organisms were yellow. In the original literature description of *Echinogorgia gracilima* (Kükenthal, 1917), the coral bodies had numerous lateral branches on a plane. The branch parts overlapped. The polyps were small. The distance between the polyps was about 1 mm, and the diameter was about 0.5 mm. The sclerites of the coral polyps were spindle-shaped with a length of 0.18 mm. The sclerites contained warty protrusions, and the coenenchymata contained leaf-shaped sclerites [51]. The morphological description of the sample with sample number 20210423-WY-11 corresponded to the original literature description of *E. gracilima* (Kükenthal, 1917); as such, it is identified as *E. gracilima* (Kükenthal, 1917).

As of 2 May 2023, WORMS (https://www.marinespecies.org, accessed on 2 March 2023) has recorded 42 valid species of this genus. After collecting and organizing the relevant original literature on forty-two species, it was found that the original literature on seven of the species could not be collected, and the original literature on nineteen of the species did not attach a species comparison map. In addition, the original literature describing the species of the genus *Echinogorgia* also had certain limitations such as simplistic morphological descriptions, a lack of corresponding example maps, and incomplete literature [20,52,53]. This made it difficult for certain samples to be identified as certain species through their morphology alone. Therefore, there are still 10 undetermined species in this study.

In addition to the corals of this genus, the identification of other genera was also the same, let alone for those at the species level. Koido et al. [54] mentioned that species identification is difficult due to the high morphological variability and plasticity of the Xeniidae family. Regarding the classification of *Chrysogorgia*, Untiedt et al. [55] believed that morphological information is still needed for taxonomic revisions. In particular, species identification relies on the observation of subtle features that are not easily accessible to non-experts. Furthermore, phenomena such as homogeneous traits, subtle differences, and environmental changing forces may blur the boundaries of coral species [56–58].

The descriptions of many octocorals still rely on morphological evidence, and an integrated approach is clearly needed but not yet prevalent [59]. Untiedt et al. [55] mentioned that an integrated approach assessing morphological and molecular variation is needed to address classification issues. Information integration is often considered the key to alleviating octocoral taxonomic problems [60–62]. The integration of genetic and morphological data is often considered key to mitigating taxonomic issues in octocorals [60–62]. As early as 2001, the scholar Bayer mentioned that new technologies in molecular analysis will allow for the possibility of solving the concept of species and genus, especially in problems that change with long-distance changes in geographical and ecological conditions [63].

### 4.2. Phylogenetic Analysis of the Genus Echinogorgia

Various molecular markers, such as mtMuts, ND2, and 28S rDNA have been used to differentiate octocoral species [64–67]. Few molecular phylogenetic analyses of xeniids have

been performed at the species level. Therefore, COI, mtMuts, ND2, and 28S rDNA were used to compare some of the genera of xeniids. However, there are still some problems and limitations [64]. Furthermore, molecular techniques require large amounts of high-quality DNA, and the need to extract it from recently collected material is critical [68,69].

Given the challenges faced in the morphological identification of certain genera of coral species in Octocorallia, many scholars have conducted phylogenetic analysis on coral species to further clarify their species relationships and to help confirm their species status. Quattrini et al. [70] established a phylogenetic tree based on mtMuts and 28S rRNA gene sequence fragments for the purposes of phylogenetic analysis, but the results were inconsistent with the morphological results. Arrigoni et al. [71] used COI gene fragments for molecular markers to study the molecular biology of coral species in the Indian Ocean; however, these also differed from the morphological results. Through the specific amplification and sequencing of COI genes, Li et al. [34] compared and analyzed the sequence of COI gene fragments of eight species of corals in Xuwen Island. They clearly pointed out that the traditional morphological classification results differed slightly from the results of their molecular phylogenetic analysis [34]. Therefore, the classification and identification of the corals required a combination of morphology descriptions and molecular biology [18,72].

(1) In this study, the phylogenetic analysis of the genus *Echinogorgia*, the COI gene sequence fragments, and the mtMuts gene sequence fragments could only be identified as a genus class, while the ITS gene sequence fragments could be identified as a species class. Nehoray et al. [73] mentioned that phylogenetic analysis of corals in the Eilat Sea area of the Red Sea using COI gene. That analysis identified 14 families, 39 genera, and 94 species. In the paper of Li et al. [74], the phylogenetic relationships of eight species of scleractinian corals were analyzed using mitochondrial COI genes. The results showed that the COI gene sequence fragments could identify the species class of the scleractinian corals, but there were certain differences in terms of morphological identifications. Zhou et al. [75] selected the mitochondrial msh1 gene and COI gene sequences for species identification when identifying the Alcyonacea of Sarcophyton in the sea area near Hainan Island. The results showed that, of the twenty-six samples collected, five species were identified. Xiao et al. [76] used ITS gene fragments to study the evolutionary relationship of the scleractinian coral systems in the outer Lingding Island, and the results showed that the ITS gene fragment could help with identifying the species class. Xie et al. [35] used the ITS gene to analyze the phylogenetic relationship of scleractinian corals in the waters of the Dapeng Peninsula. The results showed that ITS could help with identifying the species class, but there was a significant difference found in comparison with the morphological characteristics. According to current research, ITS gene fragments are mainly used for the phylogenetic analysis of coral. In this study, ITS gene fragments were used for a systematic analysis of the coral collected from Xiamen Bay, and the results showed that the ITS gene fragment could help with identifying the species class. Further research and analysis are required to test the application of ITS gene fragments in the coral species.

(2) For a phylogenetic analysis of corals of different genera, it is necessary to identify the characteristic gene fragments of that genus. With the rapid development of molecular biology, the use of DNA barcoding technology has become an important method of genetic differentiation among closely related species [77,78]. However, at present, DNA barcoding is not applicable to the identification of all coral species. In this study, only ITS gene fragments were applicable to the differentiation of the species of the genus *Echinogorgia*, while COI gene sequence fragments and mtMuts gene sequence fragments are applicable to the differentiation of *Echinogorgia*. The selection of DNA barcoding is particularly important for coral species identification.

(3) Global coral researchers are needed to jointly improve coral molecular information in the NCBI database. As of February 2023, in the NCBI database, there are thirty instances of molecular sequence data regarding the genus *Echinogorgia*, including five COI and eight mtMuts gene sequences. Most of them are the undetermined species of the genus *Echinogorgia*'s gene. In this paper, the species information of the genus *Echinogorgia* are compared and screened in the NCBI database, based on the COI gene sequence fragments. The mtMuts gene sequence fragment was used to establish a phylogenetic tree for the phylogenetic analysis of the species of the genus *Echinogorgia*, which was then validated through morphological results. Further research is needed to enrich the molecular data information of the species in the genus *Echinogorgia*.

Recently, new techniques such as the target capture enrichment of ultra-conserved elements are often considered and used [79–84]. This technology has advantages over traditional sequencing methods such as RAD-seq [55]. Erickson et al. [83] believed that new approaches to coral species delineation are needed to overcome certain challenges. Boundaries and group structure within species of two octocoral genera (Alcyium and Sinularia) were tested by using the method described above. Quattrini et al. [84] advised that target enrichment methods have shown effectiveness in resolving relatively old phylogenetic relationships. In the future, the use of target enrichment methods in terms of phylogenetic relationships may be considered and applied.

*4.3. Distribution of Species in the Genus Echinogorgia*

The species of the *Echinogorgia* genus are mainly distributed in Hong Kong, Germany, the Indian Ocean, and other such places [21]. According to certain reports, seven species of this genus have been found, including *E. pseudosassapo*, *E. ridley* [22], *E. sassapo*, *E. furfuracea*, *E. complexa*, *E. aurantiaca*, and *E. lami* in the offshore waters of China [8,11,23]. Among them, *E. pseudosasapo* and *E. ridley* inhabit the offshore waters of Taiwan, while the rest inhabit the shallow waters along the coast of Fujian. Moreover, *E. complexa*, *E. pseudosasapo*, and *E. lami* are also distributed in the waters of Hong Kong [85]. *E. pseudosasapo* is distributed in the offshore waters of Hong Kong, Taiwan and in the shallow waters of Fujian. The results show that most of the discovered species of *Echinogorgia* in China's offshore waters are distributed in the shallow waters off the coast of Fujian. However, the answer as to whether other coastal waters in China are inhabited by *Echinogorgia* species remains to be determined.

*4.4. Management Recommendations*

The global decline in coral cover has resulted in serious consequences, thereby requiring a reflection and evaluation of current management measures [86]. The health and abundance of coral cover are declining for many reasons, including overfishing, climate change, runoff pollution, plastic pollution, coastal construction, and dynamite fishing [87]. Jones et al. [88] showed that fish biodiversity is also threatened whenever permanent reef degradation occurs. From single corals (e.g., genetics, reproduction, and physiology) to coral populations, reef communities, and ecosystems, a variety of coral reef restoration approaches are currently being pursued [89]. Species identification and verification are, therefore, necessary for coral ecosystem management. This fundamental task in species identification in corals has not yet been fully scientifically described. In this study, the newly recorded species of *Echinogorgia* were revealed to have contributed to enriching the diversity of coral species in China. However, this type of meaningful information still requires attention to successfully complete complicated tasks in the future. Clearly, the work on coral population management techniques—which include the parameters of population status, as well as population dynamics and assessment—is rather limited.

Hein et al. [89] referred to the National Academies of Sciences, Engineering, and Medicine (NASEM), as well as the Reef Restoration and Adaptation Program (RRAP), as examples of some of the interventions that could improve the physiological resilience of coral to climate change [90,91]. Eleven conservation zones, with corals as the main protection targets, have been established and implemented in nature reserves in China [92]. One of them is located in the Dongshan Coral Nature Reserve in Fujian [93]. Effective conservation measures have been demonstrated in the establishment of these protected areas. Therefore, it appears advisable to establish similar coral protection areas in coral distribution and concentration areas in Xiamen Bay. According to the World Resources Institute [94] report, an estimated 2679 coral reef areas worldwide have marine protected areas (MPAs), which account for about 27% of all reef areas. Fernandez et al. [95] identified certain key success factors in protected area measures, including communication on the issues to be solved, organizing independent experts, extensive and participatory consultations, obtaining high-level support, and solving fishermen problems. To address the need for coral conservation, MPA managers need to understand why MPAs were created, the state and status of their resources, the forces that are impacting their future the most, as well as the essential management actions, equipment, people, and facilities required. Once MPAs have been established, it is important to determine how to promote and adapt the relevant management efforts [96]. Therefore, after the establishment of coral protection areas, the protection of coral species diversity in Xiamen Bay will need be implemented, including the establishment of a network of protection areas, strengthening the fishing management of fishermen, reducing land-based pollutants, and eliminating illegal coral fishing.

## 5. Conclusions

Xiamen Bay is a sea area with typical ecological characteristics and extremely high biodiversity. Understanding the diversity of coral species is crucial. This study has added the identification of 15 new species of the genus *Echinogorgia*, including *E. ramosa*, *E. flexilis*, *E. russelli*, *E. ramulosa*, and *E. gracilima*, which are new recorded species in Xiamen Bay and also in China. The morphological description and molecular analysis of these five species can also provide reference for other scholars to study the species of this genus. The results have also enriched the coral record. We believe that, based on the existing research results, government authorities should direct their attention and should design specific measures for coral protection as soon as possible in order to protect precious coral ecologies.

**Author Contributions:** Conceptualization, Y.-P.W. and J.-Y.L.; methodology, Y.-P.W. and J.-Y.L.; software, Y.-P.W., J.-Y.L., J.Y. and T.-J.C.; validation, Y.-P.W. and J.-Y.L.; investigation, J.-Y.L., Y.-P.W. and J.Y.; resources, J.-Y.L.; data curation, Y.-P.W. and J.-Y.L.; writing—original draft preparation, J.-Y.L. and Y.-P.W.; writing—review and editing, T.-J.C. and J.-Y.L.; visualization, Y.-P.W. and J.-Y.L.; supervision, J.-Y.L.; project administration, J.-Y.L.; funding acquisition, J.-Y.L. All authors have read and agreed to the published version of the manuscript.

**Funding:** This research was funded by the Natural Science Foundation of Fujian Province (No. 2019J01690), Xiamen Ocean and Fishery Bureau (Southern Center Project) (No. 13GQT001NF14), and National Foundation Incubation Program of Jimei University (No. ZP2020021).

**Data Availability Statement:** Not applicable.

**Acknowledgments:** We thank Shi Yi-Jia for her suggestions and contributions to the manuscript.

**Conflicts of Interest:** The authors declare no conflict of interest.

## Appendix A

**Table A1.** The GenBank accession numbers and reference species involved in this study.

| Number | Species | GenBank Accession Number | Gene Segment |
|--------|---------|--------------------------|--------------|
| 1 | *Echinogorgia* sp1 | OQ001251 | |
| 2 | *Echinogorgia* sp2 | OQ001252 | |
| 3 | *Echinogorgia* sp4 | OQ001253 | |
| 4 | *Echinogorgia* sp5 | OQ001254 | |
| 5 | *Echinogorgia ramosa* | OQ001255 | |
| 6 | *Echinogorgia flexilis* | OQ001256 | |
| 7 | *Echinogorgia* sp6 | OQ001257 | |
| 8 | *Echinogorgia* sp7 | OQ001258 | COI |
| 9 | *Echinogorgia* sp8 | OQ001259 | |
| 10 | *Echinogorgia* sp9 | OQ001260 | |
| 11 | *Echinogorgia* sp10 | OQ001261 | |
| 12 | *Echinogorgia ramulosa* | OQ001262 | |
| 13 | *Echinogorgia gracillima* | OQ001263 | |
| 14 | *Echinogorgia* sp. | MW401652.1 | |
| 15 | *Echinomuricea* sp. | KC984637.1 | |
| 16 | *Echinogorgia* sp1 | OQ061094 | |
| 17 | *Echinogorgia* sp3 | OQ061095 | |
| 18 | *Echinogorgia* sp4 | OQ061096 | |
| 19 | *Echinogorgia* sp5 | OQ061097 | |
| 20 | *Echinogorgia ramosa* | OQ061098 | |
| 21 | *Echinogorgia flexilis* | OQ061099 | |
| 22 | *Echinogorgia* sp6 | OQ061100 | |
| 23 | *Echinogorgia* sp7 | OQ061101 | mtMuts |
| 24 | *Echinogorgia* sp8 | OQ061102 | |
| 25 | *Echinogorgia* sp9 | OQ061103 | |
| 26 | *Echinogorgia* sp10 | OQ061104 | |
| 27 | *Echinogorgia russelli* | OQ061105 | |
| 28 | *Echinogorgia ramulosa* | OQ061106 | |
| 29 | *Echinogorgia complexa* | HQ694670.1 | |
| 30 | *Echinogorgia complexa* | HQ694668.1 | |
| 31 | *Echinogorgia* sp1 | OQ002388 | |
| 32 | *Echinogorgia* sp2 | OQ002387 | |
| 33 | *Echinogorgia* sp3 | OQ002389 | |
| 34 | *Echinogorgia* sp8 | OQ002390 | |
| 35 | *Echinogorgia russelli* | OQ002405 | ITS1 |
| 36 | *Echinogorgia ramulosa* | OQ002404 | |
| 37 | *Menella indica* | OP984327 | |
| 38 | *Euplexaura* sp2 | OQ002379 | |

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
