# Peer review of "The Species Diversity of the Genus Echinogorgia in Xiamen Bay and Its New Record in China"

_water, doi:10.3390/w15203547_

Round 1

Reviewer 1 Report

The Authors present a paper regarding specie diversity of Echinogorgia on Xiamen Bay, China.

They conducted a taxonomic research by morphological and molecular analyses and built a phylogenetic tree by DNA barcoding technology.

All the information about the populations of a specific habitat is important to understand the condition of a particular environment, its biodiversity, its possible changes and its consequent repercussions which may also have an economic impact.

The study is interesting, the Authors worked seriously and the results are worthy of note.

Some notes for the Authors:

- The introduction, although exhaustive, is not very fluent and this makes it tiring to read, it also requires a revision of the English language;

- Sentence lines 105-107 “…… was 105 used to analyze the phylogeny of the samples. Determine the specific species of coral in 106 Xiamen Bay, and provide data support for biodiversity conservation work in Xiamen Bay.” Please check it;

- Check the entire manuscript for some uppercase and homogenize the bibliography in accordance with the journal's indications; Section 4.1 of the discussion seems more like reporting results than discussing them. My suggestion is to consider combining results and discussion in future articles similar to this one; - Figures 12-14 are not very readable, the characters are too small

Extensive revision of English language for the Introduction especially

Author Response

  1. They conducted a taxonomic research by morphological and molecular analyses and built a phylogenetic tree by DNA barcoding technology.

All the information about the populations of a specific habitat is important to understand the condition of a particular environment, its biodiversity, its possible changes and its consequent repercussions which may also have an economic impact.

Answer: We are much grateful for your careful reading of our manuscript and your valuable comments and suggestions to help improve the paper.

  1. The study is interesting, the Authors worked seriously and the results are worthy of note.

Answer: We are very grateful for your recognition and appreciation.

  1. The introduction, although exhaustive, is not very fluent and this makes it tiring to read, it also requires a revision of the English language;

Answer: We have followed your comments and added and modified some paragraphs, and also added some references. We will seek revisions from the English editor to improve this article.

  1. Sentence lines 105-107 “…… was 105 used to analyze the phylogeny of the samples. Determine the specific species of coral in 106 Xiamen Bay, and provide data support for biodiversity conservation work in Xiamen Bay.” Please check it;The introduction, although exhaustive, is not very fluent and this makes it tiring to read, it also requires a revision of the English language;

Answer: We have followed your comments and modified some errors.

We fixed the “In this study, the morphological classification method was used to identify the samples of Echinogorgia collected from Xiamen Bay. Also, using molecular techniques, phylogenetic analysis was conducted with the COI gene sequence, mtMuts gene sequence and ITS gene sequence. Determine the specific species of coral in Xiamen Bay, and provide data support for biodiversity conservation work in Xiamen Bay.” Line 105-109.

  1. Check the entire manuscript for some uppercase and homogenize the bibliography in accordance with the journal's indications;

Answer: We have followed your comments and modified some errors.

  1. Section 4.1 of the discussion seems more like reporting results than discussing them. My suggestion is to consider combining results and discussion in future articles similar to this one;

Answer: We have followed your comments, and have corrected these paragraphs.

We fixed the “However, many octocoral descriptions still rely on morphological evidence, and an integrated approach is clearly needed but not yet prevalent [59]. Untiedt et al. [55] mentioned that an integrated approach assessing morphological and molecular variation is needed to address classification issues. The integration of genetic and morphological data is often considered key to mitigating taxonomic issues in octocorals [60-62]. As early as 2001, scholar Bayer mentioned that new technologies in molecular analysis will have the possibility to solve the concept of species and genus, especially in problems that change with long-distance changes in geographical and ecological conditions [63].” Line 420-427.

We fixed the “Various molecular markers, such as mtMutS, mt-cox1, igr1, ND2 and nuclear 28s rDNA have been used to differentiate octocoral species [64-66]. McFadden et al. [67] also showed that few molecular phylogenetic analyzes of xeniids have been performed at the species level. Therefore, COI, mtMutS, ND2, and 28S rDNA were used to compare Ant-helia, Cespitularia Milne Edwards & Haime, 1850, and Efflatounaria Gohar, 1939, Ovabunda, Heteroxenia, Sansibia Alderslade, 2000, and Sarcothelia Verrill, 1928 with xeniids. Furthermore, molecular techniques require large amounts of high-quality DNA, and the need to extract it from recently collected material is critical [68,69]. However, there are still some problems and limitations [64].

” Line 429-437.

We fixed the “Recently, new techniques such as target-capture enrichment of ultra-conserved elements (UCEs) and exons are often considered and used [79-84]. This technology has ad-vantages over traditional Sanger sequencing and other next-generation sequencing methods such as RAD-seq [55]. Erickson et al. [83] believe that new approaches to coral species delineation are needed to overcome some challenges. By focusing on two octo-coral genera (Alcyium and Sinularia) as exemplary case studies, they tested whether UCEs and exons can be used to define species boundaries and population structure within coral species. Quattrini et al. [84] point that results demonstrate the utility of this target-enrichment approach to resolve phylogenetic relationships from relatively old to recent divergences. The utility of this target enrichment approach in resolving phylogenetic relationships from relatively old to recent divergences was demonstrated. In the future, the use of this target enrichment method in terms of phylogenetic relationships may be considered and applied.” Line 493-505.

 7. Figures 12-14 are not very readable, the characters are too small

Answer: We have followed your comments and a new picture is being drawn, enlarge the font.

Reviewer 2 Report

The approach in the paper is interesting. The only little problem is that the sample covers only two years.

The English in the paper is good. A simple review can lead to a better readability.

Author Response

  1. The approach in the paper is interesting. The only little problem is that the sample covers only two years.Comments on the Quality of English Language:

The English in the paper is good. A simple review can lead to a better readability.

Answer: We are much grateful for your careful reading of our manuscript and your valuable comments and suggestions to help improve the paper.

Actually, our sampling is over two years. We have completed Genus Astrogorgia and published as followed:

Liu, J.-Y.; Wang, Y.-P.; Yang, J.; Shih, Y.-J.; Chu, T.-J. Revealing the Coral Species Diversity in Xiamen Bay: Spatial Distribution of Genus Astrogorgia (Cnidaria, Alcyonacea, Plexauridae) and Newly Recorded Species. Water 2022, 14, 2417.

https://doi.org/10.3390/w14152417Revealing the Coral Species Diversity in Xiamen Bay: Spatial Distribution of Genus Astrogorgia (Cnidaria, Alcyonacea, Plexauridae) and Newly Recorded Species. From 2014 to 2021, a total of 1185 samples were collected by diving in Xiamen Bay.

  1. Comments on the Quality of English Language:

The English in the paper is good.

Answer: We are very grateful for your recognition and appreciation.

  1. A simple review can lead to a better readability.

Answer: We have followed your comments and added and modified some paragraphs, and also added some references. We feel that the readability of this article has been greatly improved. 

Reviewer 3 Report

The manuscript titled "Species Diversity of the Genus Echinogorgia in Xiamen Bay and its New Record in China" intends to taxonomye species in genus Echinogorgia which belongs to the Octocorallia, Malacalcyonacea families. The morphological identification, electronic microscopy and gene fragment sequencing methods were used for taxonomic study. Diving surveys were conducted in Xiamen Bay in 2017 and 2021, and a total of 928 samples were collected. The manuscript compares the results of two years of 928 samples conducted in Xiamen Bay.

The research is original; it could be characterized as novel and in my opinion important to the field, it also has an appropriate structure, and the language has been used well. In the meanwhile, the manuscript has a good extent (about 6,100 words) and it is almost comprehensive. The tables (2), figures (14) and appendix make the paper reflect well to the reader. For this reason, paper has a "diversity look", not only tables, not only numbers, not only words. It is advised to revise figures, compare them (for example figures 8 & 9, 10 & 11 and 12, 13 & 14) or use an appendix or present some figures with tables. The total number of tables and figures is very big.

The title, I think, is all right. The abstract did not reflect well the findings of this study, and it was not the appropriate length. Please revise the abstract of the manuscript and do not forget abstract need to encourage readers to download the paper. The Abstract needs further work. It is not clear. Abstracts should indicate the research problem/purpose of the research, provide some indication of the design/methodology/approach taken, the findings of the research and its originality/value in terms of its contribution to the international literature. The abstract has a long length (about 250 words). Please, revise the abstract, it must be up to 200 words long, for this reason I would be good to reduce [see: Instructions for Authors / Manuscript Submission Overview / Accepted File Formats - (https://www.mdpi.com/journal/land/instructions#submission or https://www.mdpi.com/files/word-templates/water-template.dot)].

The introduction is effective, clear, and well organized but it wasn’t introduced and put into perspective what research is negotiating. Moreover, it does not contain a clear formulation and description of the research problem. Please insert a clear description and justification of the problem the article deals with. Your literature research should be critical and more informed, rather than listing previous research. This section requires significant improvement.

For the Methodology chapter, the research conduct has been tested in several areas of the world, with comparable results and will probably be tested in others. Appropriate references to the methodology included in the already published bibliography but you must input more.

The results section is good. The argument flows and is reinforced through the justification of the way elements are interpreted. But the same does not apply to the Discussion and Conclusion. Both sections should be consistent in terms of Proposal, Problem statement, Results, and of course, future work. Your conclusion section does not do justice to your work. Make your key contributions, arguments, and findings clearer. You must refer to the literature and previous studies in your discussion section.

You use many acronyms, but you do not give the explanation at the first use.

I believe that the conclusions section or discussion should also include the main limitations of this study and incorporate possible policy implications. I think, something more should be said about practical implications.

Please revise the references of the manuscript and include references which are already exists in bibliography. I would be much more satisfied if the number of references was slightly higher (about 25 - 30 references) and I would appreciate it if also included data from the entire world (America, Europe and Australia e.tc.). In this way it is documented that a project which is tested in a place with its own characteristics can be implemented in other places around the world. You are using too many references in Chinese.

Minor editing of English language required.

Author Response

  1. The manuscript titled "Species Diversity of the Genus Echinogorgia in Xiamen Bay and its New Record in China" intends to taxonomye species in genus Echinogorgia which belongs to the Octocorallia, Malacalcyonacea families. The morphological identification, electronic microscopy and gene fragment sequencing methods were used for taxonomic study. Diving surveys were conducted in Xiamen Bay in 2017 and 2021, and a total of 928 samples were collected. The manuscript compares the results of two years of 928 samples conducted in Xiamen Bay.

Answer: We are much grateful for your careful reading of our manuscript and your valuable comments and suggestions to help improve the paper.

  1. The research is original; it could be characterized as novel and in my opinion important to the field, it also has an appropriate structure, and the language has been used well. In the meanwhile, the manuscript has a good extent (about 6,100 words) and it is almost comprehensive. The tables (2), figures (14) and appendix make the paper reflect well to the reader. For this reason, paper has a "diversity look", not only tables, not only numbers, not only words. It is advised to revise figures, compare them (for example figures 8 & 9, 10 & 11 and 12, 13 & 14) or use an appendix or present some figures with tables. The total number of tables and figures is very big.

Answer: We have followed your comments, and have corrected these figures. These figures are all required. Therefore, we merged them to avoid verbosity.

  1. The title, I think, is all right. The abstract did not reflect well the findings of this study, and it was not the appropriate length. Please revise the abstract of the manuscript and do not forget abstract need to encourage readers to download the paper. The Abstract needs further work. It is not clear. Abstracts should indicate the research problem/purpose of the research, provide some indication of the design/methodology/approach taken, the findings of the research and its originality/value in terms of its contribution to the international literature. The abstract has a long length (about 250 words). Please, revise the abstract, it must be up to 200 words long, for this reason I would be good to reduce [see: Instructions for Authors / Manuscript Submission Overview / Accepted File Formats - (https://www.mdpi.com/journal/land/instructions#submission or https://www.mdpi.com/files/word-templates/water-template.dot)].

Answer: We have paid attention to your comments and modified the abstract. The length of the abstract has been modified to not exceed 200 words.

  1. The introduction is effective, clear, and well organized but it wasn’t introduced and put into perspective what research is negotiating. Moreover, it does not contain a clear formulation and description of the research problem. Please insert a clear description and justification of the problem the article deals with. Your literature research should be critical and more informed, rather than listing previous research. This section requires significant improvement.

Answer: We have followed your comments, and have corrected these paragraphs.

We fixed the “The places where Scleractinian corals are distributed in Fujian include Xiamen, Dongshan, Niushan Island, Taishan Islands and other islands [8,9]. However, there are currently few studies on coral classification in Xiamen Bay.” Line 53-56.

We fixed the “Octocorallia are an important component of the coral reef ecosystem, including blue corals, soft corals, sea pens, and gorgonians (sea fans, sea whips) within three orders: Alcyonacea, Helioporacea, and Pennatulacea [17].” Line 66-69.

We fixed the “Most of the above studies show the status of unspecified species [29,30], this uncertainty is not conducive to future research and production applications. Because previous authors did not specify a holotype, and illustrations of specimens and sclerites in older publications are mostly insufficient for correct species identification. In addition, some species have been described based on only a few specimens or fragments, adding to the uncertainty of species classification [31]. Therefore, an in-depth taxonomic study of species in this genus is crucial.” Line 89-95.

  1. For the Methodology chapter, the research conduct has been tested in several areas of the world, with comparable results and will probably be tested in others. Appropriate references to the methodology included in the already published bibliography but you must input more.

Answer: We've taken note of your comments and added these paragraphs and references.

We fixed the “2.4. Identification method: Generally, the taxonomic identification and description of octocorals is based on external morphology and internal morphology. These morphological characteristics include colonies size, shape and color and calyx structures, as well as sclerite content, dominance, shape, size and arrangement [31,32,37,41].” Line 134-137.

  1. The results section is good. The argument flows and is reinforced through the justification of the way elements are interpreted. But the same does not apply to the Discussion and Conclusion. Both sections should be consistent in terms of Proposal, Problem statement, Results, and of course, future work. Your conclusion section does not do justice to your work. Make your key contributions, arguments, and findings clearer. You must refer to the literature and previous studies in your discussion section.

Answer: We are very grateful for your recognition and appreciation. We have followed your comments, and very much agree with your opinion. We have enhanced the Discussion and Conclusion sections.

  1. I believe that the conclusions section or discussion should also include the main limitations of this study and incorporate possible policy implications. I think, something more should be said about practical implications.

Answer: We have followed your comments, and have corrected these paragraphs.

We fixed the “However, many octocoral descriptions still rely on morphological evidence, and an integrated approach is clearly needed but not yet prevalent [59]. Untiedt et al. [55] mentioned that an integrated approach assessing morphological and molecular variation is needed to address classification issues. The integration of genetic and morphological data is often considered key to mitigating taxonomic issues in octocorals [60-62]. As early as 2001, scholar Bayer mentioned that new technologies in molecular analysis will have the possibility to solve the concept of species and genus, especially in problems that change with long-distance changes in geographical and ecological conditions [63].” Line 420-427.

We fixed the “Various molecular markers, such as mtMutS, mt-cox1, igr1, ND2 and nuclear 28s rDNA have been used to differentiate octocoral species [64-66]. McFadden et al. [67] also showed that few molecular phylogenetic analyzes of xeniids have been performed at the species level. Therefore, COI, mtMutS, ND2, and 28S rDNA were used to compare Anthelia, Cespitularia Milne Edwards & Haime, 1850, and Efflatounaria Gohar, 1939, Ovabunda, Heteroxenia, Sansibia Alderslade, 2000, and Sarcothelia Verrill, 1928 with xeniids. Furthermore, molecular techniques require large amounts of high-quality DNA, and the need to extract it from recently collected material is critical [68,69]. However, there are still some problems and limitations [64].

” Line 429-437.

We fixed the “Recently, new techniques such as target-capture enrichment of ultra-conserved elements (UCEs) and exons are often considered and used [79-84]. This technology has advantages over traditional Sanger sequencing and other next-generation sequencing methods such as RAD-seq [55]. Erickson et al. [83] believe that new approaches to coral species delineation are needed to overcome some challenges. By focusing on two octo-coral genera (Alcyium and Sinularia) as exemplary case studies, they tested whether UCEs and exons can be used to define species boundaries and population structure within coral species. Quattrini et al. [84] point that results demonstrate the utility of this target-enrichment approach to resolve phylogenetic relationships from relatively old to recent divergences. The utility of this target enrichment approach in resolving phylogenetic relationships from relatively old to recent divergences was demonstrated. In the future, the use of this target enrichment method in terms of phylogenetic relationships may be considered and applied.” Line 493-505.

We fixed the “5. Conclusions

Xiamen Bay is a sea area with typical ecological characteristics and extremely high biodiversity. It is crucial to understand the diversity of coral species. This study has added 15 new species of the genus Echinogorgia, including E. ramosa, E. flexilis, E. russelli, E. ramulosa, E. gracilima, which are new recorded species in Xiamen Bay and even in China. The morphological description and molecular analysis of these five species can also provide reference for other scholars to study species of this genus. The results of this study enrich the record of species diversity. We believe that based on the existing research results, government authorities should pay attention and design specific measures for coral protection as soon as possible to protect the precious coral ecology.” Line 556-564.

  1. Please revise the references of the manuscript and include references which are already exists in bibliography. I would be much more satisfied if the number of references was slightly higher (about 25 - 30 references) and I would appreciate it if also included data from the entire world (America, Europe and Australia e.tc.). In this way it is documented that a project which is tested in a place with its own characteristics can be implemented in other places around the world. You are using too many references in Chinese.

Answer: We have followed your comments, and have added 25 references. We are very grateful for your advice. It makes this article completer and more readable.

Round 2

Reviewer 3 Report

The manuscript titled "Species Diversity of the Genus Echinogorgia in Xiamen Bay and its New Record in China" intends to taxonomye species in genus Echinogorgia which belongs to the Octocorallia, Malacalcyonacea families. The morphological identification, electronic microscopy and gene fragment sequencing methods were used for taxonomic study. Diving surveys were conducted in Xiamen Bay in 2017 and 2021, and a total of 928 samples were collected. The manuscript compares the results of two years of 928 samples conducted in Xiamen Bay.

The manuscript has been revised according to the review comments. The authors carefully studied the comments and revised the manuscript by considering all the last comments. The comments are responded to the new manuscript.

Conclusions and discussion are better than the previous one, they have general logic and on justification of interpretations as the author’s attribute.

In general, the manuscript is completely different from the previous one, since all the comments of the review have been revised. The number of references is higher.

I believe the revised manuscript has been improved carefully and I hope the desired level of Water can be reached.

Minor editing of English language required.